# Moderate-High Disease Activity in Patients with Recent-Onset Psoriatic Arthritis—Multivariable Prediction Model Based on Machine Learning

**DOI:** 10.3390/jcm12030931

**Published:** 2023-01-25

**Authors:** Rubén Queiro, Daniel Seoane-Mato, Ana Laiz, Eva Galindez Agirregoikoa, Carlos Montilla, Hye S. Park, Jose A. Pinto Tasende, Juan J. Bethencourt Baute, Beatriz Joven Ibáñez, Elide Toniolo, Julio Ramírez, Nuria Montero, Cristina Pruenza García-Hinojosa, Ana Serrano García

**Affiliations:** 1Rheumatology Service & the Principality of Asturias Institute for Health Research (ISPA), Faculty of Medicine, Universidad de Oviedo, 33006 Oviedo, Spain; 2Research Unit, Spanish Society of Rheumatology, 28001 Madrid, Spain; 3Rheumatology and Autoimmune Disease Department, Hospital Universitari de la Santa Creu i Sant Pau, 08025 Barcelona, Spain; 4Rheumatology Service, Hospital Universitario Basurto, 48013 Bilbao, Spain; 5Rheumatology Service, Hospital Universitario de Salamanca, 37007 Salamanca, Spain; 6Rheumatology Service-INIBIC, Complexo Hospitalario Universitario de A Coruña, 15006 A Coruña, Spain; 7Rheumatology Service, Hospital Universitario de Canarias, 38320 Sta. Cruz de Tenerife, Spain; 8Rheumatology Service, Hospital Universitario 12 de Octubre, 28041 Madrid, Spain; 9Rheumatology Service, Hospital Universitari Son Llàtzer, 07198 Palma de Mallorca, Spain; 10Arthritis Unit, Rheumatology Department, Hospital Clínic Barcelona, 08036 Barcelona, Spain; 11Knowledge Engineering Institute, Universidad Autónoma de Madrid, 28049 Madrid, Spain

**Keywords:** arthritis, psoriatic, disease activity, DAPSA, predictive model, machine learning, PsAID

## Abstract

The aim was to identify patient- and disease-related characteristics predicting moderate-to-high disease activity in recent-onset psoriatic arthritis (PsA). We performed a multicenter observational prospective study (2-year follow-up, regular annual visits) in patients aged ≥18 years who fulfilled the CASPAR criteria and had less than 2 years since the onset of symptoms. The moderate-to-high activity of PsA was defined as DAPSA > 14. We trained a logistic regression model and random forest–type and XGBoost machine learning algorithms to analyze the association between the outcome measure and the variables selected in the bivariate analysis. The sample comprised 158 patients. At the first follow-up visit, 20.8% of the patients who attended the clinic had a moderate-to-severe disease. This percentage rose to 21.2% on the second visit. The variables predicting moderate-high activity were the PsAID score, tender joint count, level of physical activity, and sex. The mean values of the measures of validity of the machine learning algorithms were all high, especially sensitivity (98%; 95% CI: 86.89–100.00). PsAID was the most important variable in the prediction algorithms, reinforcing the convenience of its inclusion in daily clinical practice. Strategies that focus on the needs of women with PsA should be considered.

## 1. Introduction

Psoriatic arthritis (PsA) is a chronic inflammatory disease included in the group of spondyloarthritis, although the multifaceted nature of this entity has led it to be addressed in a differentiated and specific manner [1]. According to recent data, the prevalence of PsA in adults is approximately 0.6%, thus making this disease one of the most common in rheumatology clinics [2].

Evaluation of the activity of PsA has been mostly directed towards developing monodimensional instruments, such as the DAPSA (Disease Activity Index for Psoriatic Arthritis), which is a simple sum of the number of painful joints, the number of swollen joints, the pain, and the general estimate that the patient makes about his or her disease, to which the C-reactive protein value can be added (optional). However, given the multifaceted profile of the disease, multidimensional instruments have also been developed to ensure a broader assessment of patients [3]. The latter is focused on capturing the multiple facets of the disease (skin, joints, physical function, quality of life, etc.) as opposed to the one-dimensional instruments designed only to assess the musculoskeletal component. The European League Against Rheumatism (EULAR) favors the former in its recommendations on the management of PsA, whereas the Group for Research and Assessment of Psoriasis and Psoriatic Arthritis clearly favors the latter [3,4].

In any case, an area of constant doubt requiring further research is that of identifying which characteristics of PsA can predict greater disease activity and/or severity. Traditional recommendations by EULAR on the management of PsA state the factors indicative of a poor prognosis to be female sex, elevated acute phase reactant values, dactylitis, early structural damage, and a high number of joints involved [4]. Nevertheless, it is difficult to discern whether these and other factors reported in the literature refer more to disease activity itself (inflammatory burden), cumulative damage (functional burden), or the overall impact of the disease, which brings together the previous factors. Indeed, the concept of severity should not be made homonymous with that of inflammatory load. The severity of the disease brings together not only the inflammatory load of the disease but also physical disability, structural damage, and the impact on quality of life, among others. The factor that perhaps weighs the most in this equation, especially at the beginning of the disease, is inflammatory activity, but with the passage of time, this loses its predictive capacity, and there are other aspects (structural damage, comorbidities, deterioration in physical function, etc.) more clearly associated with greater severity [3,4,5,6,7]. In addition, more than one study has shown how the outcome measures on disease activity and those on the negative impact on QoL are clearly discordant, which suggests that both aspects measure different things within the overall health of patients with PsA [3,4,5,6,7].

Therefore, being able to predict which aspects of PsA are associated with increased disease activity is not only essential for better management of the disease but also an unresolved issue. From a clinical practice viewpoint, the need for information on predictors of greater disease activity is immediate, given that a suitable approach to inflammatory activity could limit or prevent structural damage.

The objective of the present study was to evaluate which patient- and disease-related characteristics predict moderate-to-high disease activity in patients with recent-onset PsA during the first years of follow-up.

## 2. Materials and Methods

This work is part of the REAPSER study, a multicenter observational prospective study with a 2-year follow-up and regular annual visits, promoted by the Spanish Society of Rheumatology. The design of REAPSER has been described in detail elsewhere [5,6,7,8]. Patients of both sexes, aged ≥18 years, who met the Classification Criteria for Psoriatic Arthritis (CASPAR) [9] and had less than 2 years since the onset of symptoms attributable to the disease were included.

A baseline visit evaluated the patient’s situation before disease progress was modified by the treatments prescribed by the rheumatologist. According to this, the time from treatment initiation to the baseline visit could not be longer than 3 weeks for methotrexate, leflunomide, or apremilast, and patients could not be on biologic disease-modifying antirheumatic drugs (DMARDs). These time gaps were defined according to the mean time between treatment initiation and the onset of the response (4 weeks for synthetic DMARDs and 1 week for biologic DMARDs). There were 9 patients with more than 3 weeks on synthetic DMARDs but less than 2 months; the investigating rheumatologists confirmed that these patients had not responded to treatment at the baseline visit.

Psoriatic patients treated with synthetic or biologic DMARDs that were referred to the rheumatologist because of recent onset PsA could be included in the study since this would not go against the criterion explained in the previous paragraph.

Patients were invited to participate consecutively when visiting the rheumatologist. Baseline visits took place between November 2014 and October 2016. 25 centers covering a wide area of Spain participated in the study.

Written informed consent was obtained from all participants. The study complies with the Declaration of Helsinki and was approved by the Clinical Research Ethics Committee of the Principality of Asturias (study number 14/2014).

### 2.1. Variables and Measurement

Variables included in REAPSER have been previously described [8]. For this work, we considered 43 variables:(a)Sociodemographic data: age; sex; educational level (none, primary, secondary, university).(b)Family history of PsA, other types of inflammatory arthritis, and psoriasis.(c)Comorbidities (based on a review of medical records): age-adjusted Charlson comorbidity index [10], cardiovascular risk factors (arterial hypertension, hyperlipidemia, diabetes mellitus (insulin and non–insulin-dependent)).(d)Anthropometric data: body mass index (BMI).(e)Lifestyle: smoking, alcohol consumption [11], and physical activity (low, moderate, and high) [12].(f)The clinical situation at diagnosis of PsA: year of presentation of symptoms; clinical form (axial, peripheral, mixed); articular pattern (oligoarticular, polyarticular, distal, mutilans, spondylitis); the presence of dactylitis (yes/no).(g)Joint involvement and enthesitis: number of tender joints (NTJ68); the number of swollen joints (NSJ66); an extended version of the Maastricht Ankylosing Spondylitis Enthesitis Score (MASES) [13]. Polyarthritis was defined as NSJ66 ≥5.(h)Pain and global assessment of disease during the previous week: Global pain on a scale from 0 (no pain) to 10 (very intense); patient global assessment of disease (from 0 -feels very well- to 10 -feels very ill-); physician global assessment of disease (from 0 -minimal activity- to 10 -maximum activity-).(i)Cutaneous and nail involvement (evaluated by a dermatologist): cutaneous psoriasis (yes/no); year of onset of psoriasis; clinical type; specific locations; treatment of psoriasis at PsA diagnosis. Psoriasis Area and Severity Index (PASI) [14]; onychopathy (number of digits affected). Severe psoriasis was defined as PASI >10 [14].(j)Functional situation and quality of life: Health Assessment Questionnaire (HAQ) [15], Psoriatic Arthritis Impact of Disease (PsAID) [16].(k)Radiographic evaluation at baseline: Bath Ankylosing Spondylitis Radiology Index (BASRI) of the sacroiliac region [17], hand involvement according to the modified Steinbrocker method for PsA [18].(l)Laboratory tests: C-reactive protein (CRP), uric acid, total cholesterol, LDL cholesterol, and triglycerides. For purposes of the analysis, a series of cut-off points were established to define high values: >0.5 mg/dL for standard CRP; >0.3 mg/dL for high-sensitivity CRP; hyperuricemia if >7 mg/dL in men and >6 mg/dL in women; ≥200 mg/dL for total cholesterol; ≥100 mg/dL for LDL; ≥150 mg/dL for triglycerides.(m)Treatment of PsA with DMARDs: synthetic DMARDs, biologic DMARDs (the different drugs within these two groups were not considered individually).(n)The moderate-to-high activity of PsA (binary outcome variable) is defined as DAPSA >14 [19]. DAPSA is calculated as the sum of NTJ68, NSJ66, CRP (mg/dL), patient global assessment of disease, and global pain [19].

Rheumatologists assessing the patients did not know the objectives of this specific analysis.

### 2.2. Sample Size

The REAPSER study was a cohort intended to collect a large number of variables without prespecified hypotheses, so a sample size was not previously calculated for this work.

### 2.3. Statistical Analysis

#### 2.3.1. Imputation of Missing Data:

-The duration of psoriasis was imputed with the median in the remaining patients within the same age range (<41 years, 41–60 years, and >60 years).-Since during data monitoring we observed that cases in which systemic treatment of psoriasis was not available were patients with no treatment or topical treatment, systemic treatment was imputed with 0 (not receiving it). This imputation affected only two cases.-Radiological involvement of the hands was not imputed, except for those patients with both NTJ28 and NSJ28 values of 0, who were imputed with 0.-For patients who stopped attending the visits due to improvement in their condition, the missing values for the variables PsAID, HAQ, and moderate or severe disease activity were imputed with 0.

#### 2.3.2. Generation of the Dataset

The analysis aimed to estimate predictive ability, seeking associations between the outcome measure (moderate-high activity of PsA) and values at the previous visit for the remaining variables. To do so, the dataset contained data for the independent variables from the baseline visit and from follow-up visit number 1, which were matched with the outcome measure from follow-up visits 1 and 2, respectively. Atemporal variables (such as sex) and those only collected at the baseline visit (e.g., clinical form at diagnosis) were matched with the outcome measure from follow-up visits 1 and 2; therefore, their values are the same for each one.

#### 2.3.3. Bivariate Analysis

To identify informative variables, we used two different methods: significant Spearman correlation according to the threshold applied to the p correlation coefficient ($\left|p\right|>\frac{2}{\surd N}$, with *N* being the number of data items), and methods based on artificial intelligence, specifically the XGBoost algorithm with the SHAP technique (the twenty most important variables according to this technique were identified; see Appendix A for a detailed explanation of this approach).

After informative variables were identified, we applied the Mann-Whitney test for continuous or discrete variables and the χ^2^ test for categorical variables to select those variables associated with the outcome measure (*p* < 0.05).

#### 2.3.4. Multivariate Analysis

Considering that the dataset contained data from different annual visits for the same subjects, we included the number of visits (the different time points) as a variable in the multivariate analysis, so the association of the other variables with the outcome measure is adjusted for this variable.

To generate models where the independent variables do not share information and have a significant contribution to the model when adjusting for the rest of the variables included, we identified those variables with *p* < 0.05 in an iterative fashion using logistic regression models based on artificial intelligence. To avoid overfitting, the steps were performed in 75% of the sample (the training dataset, which was generated in a random and balanced manner, i.e., the categories of the outcome measure were equally distributed in that 75% of the sample and the remaining 25%) (see Appendix A for a detailed explanation).

Subsequently, random forest–type and XGBoost machine learning algorithms were used to analyze the association between the outcome measure and the variables selected (see Appendix A for more detail). To train the machine learning models, the sample is randomly split into two subsets: one to train the model and the other to evaluate its functioning. The split is also generated in a balanced way; the categories of the outcome measure are equally distributed in both subsets. When the subsamples are imbalanced, those data whose value for the outcome variable is a minority value are duplicated or triplicated to train the models (oversampling technique).

As different splits might result in different models, k-fold cross-validation was used to reduce this randomness. It consists in splitting the dataset into k subsets of the same size, and iteratively training the algorithm with k-1 of them while testing the model with the one left. After k iterations, the algorithm will have been trained and evaluated with all the partitions. In this analysis, a k-fold cross-validation with k = 5 was performed (models were trained with 80% of the data at each iteration and then evaluated with the remaining 20%). The subsets were the same for the random forest and XGBoost.

The contribution of the variables to the prediction of each iteration of the algorithms was calculated by the feature importance of each variable in the training data. To estimate the validity of the algorithms, we calculated the accuracy, sensitivity, specificity, positive predictive, and negative predictive values as the mean of the values obtained for each parameter in the five evaluations performed in the cross-validation.

Data analysis was performed with Python (3.8.12 version, Python Software Foundation, Fredericksburg, Virginia, USA), using open-source libraries: pandas 1.3.4, numpy 1.19.0, scikit-learn 1.0, scipy 1.5.2, and statsmodels 0.13.0.

## 3. Results

The sample comprised 158 patients. The baseline characteristics have been previously published [6]; 59.6% of the patients had moderate-to-high disease activity; their rheumatologists prescribed synthetic DMARDs to 60.8% of the patients and biologic DMARDs to 1.3%.

Thirty-three patients (20.9%) were lost to follow-up. For 10 of these patients, the investigating rheumatologists could confirm that they had not attended the visit because their PsA had improved.

20.8% and 21.2% of the patients who attended the clinic had moderate-high disease activity at the first and second follow-up visits, respectively.

### 3.1. Bivariate Analysis

Table 1 shows the variables selected in the bivariate analysis.

### 3.2. Multivariate Analysis

The number of observations for this analysis was 319.

Table 2 shows the results of the logistic regression. The variables predicting moderate-high disease activity selected in this analysis were PsAID, tender joint count, level of physical activity, and sex.

When the random forest–type and XGBoost machine learning algorithms were trained with these 4 variables, PsAID was the most important variable according to the values of feature importance in all the models (Table 3).

Table 4 shows the mean of the values of accuracy, sensitivity, specificity, positive predictive value, and negative predictive value in the different evaluations performed in the cross-validation.

## 4. Discussion

In this multicenter prospective study carried out in patients with recent-onset PsA, assessed at baseline before the potential modification of its natural history because of the treatment prescribed by a rheumatologist, an artificial intelligence-based analysis revealed 4 variables that could predict moderate-to-severe disease activity (according to DAPSA, a one-dimensional tool to assess the inflammatory activity of PsA in clinical routine) on the following annual visit during the first two years after diagnosis: PsAID, tender joint count, level of physical activity, and sex. The mean values of the measures of validity of the machine learning algorithms were all high, especially sensitivity and NPV.

PsAID, a 12-item questionnaire designed to capture the impact of PsA on a patient´s daily life, was the most important variable in all the models. It is currently the gold standard for estimating the impact of PsA on a patient´s health [20]. This 12-item instrument evaluates various aspects, such as pain, skin, physical function, sleep, psychosocial aspects, the ability to work, and the ability to enjoy leisure activities. PsAID could serve to evaluate not only disease activity but also functioning and aspects that do not necessarily reflect disease activity or inflammatory burden [20]. Therefore, to a certain extent, PsAID could be an all-in-one tool for the evaluation of PsA in clinical practice. Nevertheless, analysis of the metric characteristics of PsAID with respect to other evaluation tools or disease outcomes shows that the degree of agreement ranges from discrete to considerable [21,22,23]. Furthermore, PsAID shows good sensitivity to change in real-world studies on PsA [24]. Similarly, although the debate between the use of mono and multidimensional activity indices continues in PsA, a recent multicenter study showed better agreement between a low-impact PsAID and remission according to DAPSA (k: 0.58) than with very low disease activity (VLDA response) (k: 0.18) [22]. In summary, the finding of PsAID as an independent predictor of higher disease activity as measured by DAPSA in our study seems to indicate that when patients make a global assessment of their state and record this information using DAPSA, they could be already including, albeit indirectly, information on other types of involvement (e.g., axial, enthesis, or skin).

The tender joint count was the second variable in order of importance according to the results of the random forest algorithms. Based on the results of the bivariate and logistic regression analyses, the tender joint count was more closely associated with moderate-high disease activity than polyarthritis. We should remember that many patients with PsA and no obvious arthritis in the physical examination present with subclinical inflammation—both in joints and in entheses—when sensitive imaging techniques such as high-resolution ultrasound and magnetic resonance imaging are used [25]. Inflammatory activity according to ultrasound or magnetic resonance imaging in the joints of patients in clinical remission acts as a predictor of clinical flares in the short term (6 months) and of subsequent structural damage [25]. Therefore, it is possible that some PsA patients with joint pain and no apparent inflammatory activity do, in fact, have some subclinical inflammatory activity that can only be detected using sensitive imaging techniques. This result of our analysis would not be affected by the presence of fibromyalgia, as only 1 patient in the baseline visit and 2 patients in the first follow-up visit were diagnosed with this disease. As a practical corollary to our finding, patients with apparently good control of PsA who develop joint pain should be actively screened for signs of subclinical inflammation using sensitive imaging techniques and start appropriate therapy.

We detected an inverse relationship between physical activity and higher disease activity, according to DAPSA. It is known that a session of moderate physical exercise lasting 20 to 30 min can reduce the number of TNFα-releasing immune cells by 5% [26,27]. Furthermore, though intense and acute physical exercise increases IL-6 levels, IL-6 downregulates several proinflammatory cytokines such as TNFα, helping to improve glucose metabolism and increase insulin sensitization to insulin [26,27]. In summary, it seems that physical exercise has a direct anti-inflammatory effect that contributes to the patient’s general well-being and to combating insulin resistance and obesity. The connection between obesity and inflammation is well documented [28]. Therefore, another clearly practical approach is to advise and encourage patients with PsA to adopt cardio-healthy lifestyles, since this could prevent not only the adverse cardiovascular events that are so common in PsA but possibly the development of a more active disease.

The last factor associated with greater disease activity, according to DAPSA, was female sex. Women with spondylarthritis or PsA usually have greater disease activity (as also shown by our results), a higher degree of pain (a component of DAPSA), a higher impact of the disease, and poorer persistence of biologics [29]. The association between sex and disease activity on the following year´s visit is observed after adjusting for the tender joint count and PsAID in the logistic regression analysis, pointing out that this association would go beyond different disease activity or impact in the previous visit. Women with PsA perceive greater disease impact and activity, probably because of multifactorial reasons that are beyond the scope of this study; therefore, we should make every effort to design strategies that focus on the needs of women with PsA [30].

In this work, we have described predictors of moderate-high disease activity according to the DAPSA score. We had previously described severity predictors using machine learning algorithms, as in the present work [6]. In any case, it is necessary to highlight that activity and severity are not the same concept. Inflammatory activity can be part of what we understand as disease severity, but the latter concept is much broader and encompasses other aspects of PsA that go beyond the inflammatory load, which is what DAPSA collects. Therefore, we think that the information provided in this study should be understood as complementary to our previous work and not as a circular argument.

The main limitation of this work is its sample size and missing data for some variables. This could have influenced the power of the analysis and, as a consequence, the capacity to identify variables associated with moderate-to-severe disease activity. We tried to counteract this by using models based on artificial intelligence and machine learning. Random forests are “joint” algorithms in which decision trees are trained with different subsets of variables and data. Decision trees are very flexible since they can identify many types of associations between variables. Additionally, random forests add variability, which prevents the model from being overadjusted to the data and can be re-run with new data, increasing the robustness of the predictions. On the other hand, XGBoost algorithms use groups of decision trees in a sequential manner. In each tree, the observations that were wrongly classified in the previous one are given a larger weight, thus defining models with very little bias that commonly generate very accurate predictions. The risk of this phenomenon is a higher probability of the model being overfit to the training subset. Our results showed that the random forest models tended to perform better than XGBoost, which could be a consequence of the reduced number of observations in the dataset, causing the training and test subsets to be quite different. Consequently, we could conclude that for small datasets, an algorithm that overfits less to the training subset, such as random forest, is more suitable.

Due to the relatively small sample size and short follow-up period, independent replication of our results becomes very important.

The main strength of this study is its ability to record the course of PsA from an early phase; the baseline visit reflected the situation of the patients before the evolution was modified by treatment prescribed by the rheumatologist.

Part of the interest of our work is ensuring that there is no confounding effect affecting the associations found between the explanatory variables and moderate-to-high disease activity. According to this, the four final predictors have an association with disease activity once the effect of the rest of the variables initially selected in the bivariate analysis has been taken into account. Moreover, the observed association of each of the four predictors with disease activity in the following annual visit is adjusted for the other three variables. This would reflect an association that would go, at least in part, beyond disease activity at the present moment.

## 5. Conclusions

Our artificial intelligence–based analysis revealed that the PsAID score, tender joint count, physical exercise, and female sex could predict greater disease activity according to the DAPSA index during the first years of follow-up with very high sensitivity. Once again, PsAID has shown to be important in the evaluation of PsA, reinforcing the convenience of its inclusion in daily clinical practice. Strategies that focus on the needs of women with PsA should be considered.

## Figures and Tables

**Table 1 jcm-12-00931-t001:** Variables associated with moderate-high activity (DAPSA > 14): Bivariate analysis.

Variable	DAPSA ≤ 14 (*n* = 197)	DAPSA > 14 (*n* = 62)	*p* Value ^1^
Sex (women)	78 (39.6)	37 (59.7)	0.009
Educational level			0.04
None	1 (0.5)	3 (4.8)	
Primary	72 (36.5)	28 (45.2)	
Secondary	84 (42.6)	20 (32.3)	
University	40 (20.3)	11 (17.7)	
Physical activity			0.04
Low	30 (17.8)	17 (33.3)	
Medium	84 (49.7)	23 (45.1)	
High	55 (32.5)	11 (21.6)	
No. of tender joints	1.5 (0.5–4)	6 (1.5-11)	<0.001
Polyarthritis	26 (11.4)	17 (27.4)	0.002
Global pain	3 (1.5–5.5)	6 (4.5–7)	<0.001
Physician global assessment	2.5 (1–4)	4 (2.5–6)	<0.001
PsAID score	1.64 (0.75–4.01)	4.69 (3.77–6.37)	<0.001
HAQ score	0.19 (0–0.69)	0.82 (0.42–1.32)	<0.001

Categorical variables are expressed as *n* (%) and numeric variables as median (interquartilic range). ^1^ Mann-Whitney test for numeric variables and χ^2^ test for categorical variables.

**Table 2 jcm-12-00931-t002:** Variables associated with moderate-high activity selected in the logistic regression analysis.

Variable	Regression Coefficient ^1^	95% CI	*p* Value (Wald Test)
PsAID	4.399	(2.888, 5.909)	<0.001
Tender joint count	3.618	(0.813, 6.423)	0.011
Level of physical activity	−0.913	(−1.819, −0.007)	0.048
Female sex	0.720	(0.098, 1.341)	0.023
Visit number	1.112	(0.383, 1.841)	0.003

^1^ Positive values indicate a positive direction of the association (the higher the value of the variable, the higher the frequency of moderate-high activity).

**Table 3 jcm-12-00931-t003:** Feature importances * of the variables in the different models trained in the cross validation.

Variable	Iteration 1	Iteration 2	Iteration 3	Iteration 4	Iteration 5
Random Forest					
PsAID	0.601	0.634	0.564	0.563	0.610
Tender joint count	0.248	0.233	0.286	0.290	0.247
Level of physical activity	0.055	0.067	0.060	0.057	0.073
Female sex	0.063	0.040	0.044	0.058	0.038
Visit number	0.034	0.027	0.046	0.032	0.033
XGBoost					
PsAID	0.281	0.352	0.297	0.308	0.434
Tender joint count	0.161	0.147	0.132	0.159	0.164
Level of physical activity	0.142	0.199	0.222	0.183	0.144
Female sex	0.196	0.172	0.196	0.170	0.154
Visit number	0.220	0.129	0.153	0.180	0.104

* Values from 0 to 1. The bigger the value, the bigger the importance of the variable in the model. Values are normalized, i.e., in each iteration the sum of the values equals 1.

**Table 4 jcm-12-00931-t004:** Measures of validity in the different evaluations performed in the cross validation.

Metric	Accuracy	Sensitivity	Specificity	NPV	PPV
Random Forest					
Mean *	89.97	98.00	82.51	98.13	83.93
SD	2.55	4.00	4.51	3.75	3.30
95% CI	82.88, 97.05	86.89, 100.00	70.00, 95.02	87.71, 100.00	74.77, 93.08
XGBoost					
Mean ^&^	84.31	94.75	74.65	94.47	78.22
SD	3.82	3.94	9.97	3.63	6.37
95% CI	73.71, 94.90	83.82, 100.00	46.98, 100.00	84.38, 100.00	60.53, 95.92

SD: standard deviation. * Mean of the values obtained in the 5 evaluations performed in the cross validation in Random Forest analysis. ^&^ Mean of the values obtained in the 5 evaluations performed in the cross validation in XGBoost analysis.

## Data Availability

The data presented in this study are available on request from the corresponding author.

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
