# Peer review of "Moderate-High Disease Activity in Patients with Recent-Onset Psoriatic Arthritis—Multivariable Prediction Model Based on Machine Learning"

_jcm, 2023, doi:10.3390/jcm12030931_

Round 1

Reviewer 1 Report

This manuscript by Queiro and colleagues addresses an important question: how best to predict individualised disease prognosis in order to enable early intervention. The study is based on a prospective and deeply phenotyped dataset of PsA patients that clearly has great potential for the exploration of personalised medicine approaches. The stated aim, and the results presented here, concern the identification of the most important baseline factors for prediction of subsequent moderate-high disease activity. However, I have some concerns about the approach.

Major comments:

Regarding the study design, I have concerns that the inclusion of data points from the same individuals at different time points can lead to bias and overly favourable prediction results. As far as I can tell they are not handled using a longitudinal prediction framework but are treated as independent observations by the models. My intuition is that this is not appropriate and if it cannot be carefully justified I would recommend additional steps to ensure the results are valid. E.g. can the repeat measures be explicitly accounted for? Can the analysis be performed within different timepoints separately?

Given the small dataset, it is not clear why the authors aim to predict a dichotomised DAPSA outcome when the actual quantified values are known. This feels like it would be discarding useful information. Why not build a prediction model for the actual DAPSA score? Even if the ultimate aim is for a binary predictor, it may be more effective to infer this from a score prediction.

The manuscript does not focus on the strength of the predictions themselves, perhaps because of the relatively small sample size available. Instead, it focuses on which predictors are most informative. I struggle to understand the logic of how the machine learning approaches are being deployed. It appears that a long list of variables is narrowed down to a small number using logistic regression only, and then random forest or XGBoost are being used to estimate predictor importance. However, I always understood the strengths of these ML methods to be the other way around. RF works well at generating predictions when there are a large number of potential predictors that could have non-linear, complex and interacting effects; on the other hand, prediction accuracy is being optimised and it is not well suited to interpretability. Why then constrain the variable list so heavily before applying RF?

On the interpretation of results, my takeaway is that three of the four final predictors of moderate-severe disease at follow-up are markers of more severe disease at baseline (PsAID, tender joint count, and physical activity which could be a consequence of disease level). I’m not sure how beneficial this is? Have the authors considered change in disease activity as an outcome?

It is unclear whether treatments are included as potential predictors with variable categories for individual drugs or just for class of drug. I think it’s important to give some more detail about treatment in particular so the reader can understand whether the study is predicting prognosis or treatment response. The end of the discussion states “The main strength of this study is its ability to record the course of PsA from an early phase before the natural disease evolution is modified by treatment prescribed by the rheumatologist” but presumably the effect of treatment is present in the outcome.

The authors have apparently previously published a severity prediction study in the same dataset (reference 6). This should be described more clearly in the introduction and it should be made clear what is the novel additional contribution of the current manuscript. Also worth explaining DAPSA in the introduction.

Minor comments:

** Introduction **

Line 47 – it would be beneficial to expand on what is meant by multidimensional instruments

** Methods **

More specific detail should be given. For example:

-       A supplementary table listing treatment numbers

-       Summary of the times between visits (i.e. prediction interval)

It is not clear how the initiation and finalisation dates of treatment (variable m) are handled if treatment is not stopped. Also does this include historical treatments?

It would be useful to give the total number of predictive features considered as the list is very long and it is unclear.

Also, line 131 – presumably variable (n) is the outcome variable. It should be clearer that this is the outcome and treated as a binary outcome. Or is it also included as a predictor at baseline?

Line 165 – should be XGBoost algorithm “with” the SHAP technique? (rather than “and”), i.e. one approach is being described not two? You should also be specific about the inclusion criteria for informative variables that is being applied with this approach.

What does SHAP stand for?

Line 173 – 75% training dataset – worth describing in section 2.3.2 what this is and how it is generated e.g. is it stratified/balanced in any way? Line 177 – “sample is split” – is this the same 75/25% split? It needs to be clearer.

As currently presented the 25% test dataset does not appear to be used anywhere. Please clarify

** Results **

Line 200 - 59.6% of pts have moderate-high disease activity – this is insufficiently clear in the context of multiple outcome measures per patient

Table 1 – what do table values represent? Count (%) and mean (range) or median (IQR) etc? It would benefit to add the total n for the two columns. Also perhaps indicating via footnotes which tests were performed to generate the p-values

Line 211 – 319 observations in total. Please clarify how this can be more than 158 patients x 2 visits = 316

Table 3 – I feel this can be summarised more clearly as a figure or by giving total/average importance measures

** Discussion **

Lines 259-263 – I’m not clear what point is being made. Perhaps worth reminding the reader of the differences between PsAID and DAPSA?

Line 267 – “many patients with PsA and no obvious arthritis in the physical examination” – I’m not a clinician so I’m not clear what this means – psoriatic arthritis without arthritis?

Line 275 – “have some subclinical inflammatory activity that can only be detected using sensitive imaging techniques.” – how do you infer this when your final model is based on tender joint count and not imaging data?

Physical activity levels are discussed as a causal factor of activity levels and not considered a potential consequence.

Sex as a predictor is interesting. However, it’s not clear whether this just represents “high activity predicts high activity” again, depending on whether females have higher activity at baseline. Could this test be done at baseline? I also feel that an opportunity is missed to explore further the impact of sex in relation to the other predictors, e.g. looking at potential interactions between sex and the severity-related measures in prediction of future activity.

Limitations don’t mention the lack of independent replication which I think is also key.

** Appendix A **

Line 395 – “the null hypothesis of this test is that there would be no statistically significant differences in the functioning of the model if the coefficient of the variable was zero” – a null hypothesis can’t refer to statistically significant differences. The null hypothesis is just that the effect of a variable is zero.

Is the logistic regression approach referred to simply stepwise backward regression? Why not state this?

Author Response

Major comments:

Regarding the study design, I have concerns that the inclusion of data points from the same individuals at different time points can lead to bias and overly favourable prediction results. As far as I can tell they are not handled using a longitudinal prediction framework but are treated as independent observations by the models. My intuition is that this is not appropriate and if it cannot be carefully justified I would recommend additional steps to ensure the results are valid. E.g. can the repeat measures be explicitly accounted for? Can the analysis be performed within different timepoints separately?

We appreciate the reviewer´s comment. Following the advice, we have repeated the analysis adding the number of the visit (the different time points) as a variable, so the association of the other variables with the outcome measure is adjusted for this new variable. This has not significantly modified the results of our analysis. We have made the corresponding changes in Materials and Methods (first paragraph in 2.3.4. Multivariate analysis) and in Results (3.2. Multivariate analysis).

Given the small dataset, it is not clear why the authors aim to predict a dichotomised DAPSA outcome when the actual quantified values are known. This feels like it would be discarding useful information. Why not build a prediction model for the actual DAPSA score? Even if the ultimate aim is for a binary predictor, it may be more effective to infer this from a score prediction.

Categorization of disease activity according to DAPSA values has been previously defined on an international expert survey, published in 2016 (reference 19 in our article). We have used this cut-off value as it is internationally accepted.

The manuscript does not focus on the strength of the predictions themselves, perhaps because of the relatively small sample size available. Instead, it focuses on which predictors are most informative. I struggle to understand the logic of how the machine learning approaches are being deployed. It appears that a long list of variables is narrowed down to a small number using logistic regression only, and then random forest or XGBoost are being used to estimate predictor importance. However, I always understood the strengths of these ML methods to be the other way around. RF works well at generating predictions when there are a large number of potential predictors that could have non-linear, complex and interacting effects; on the other hand, prediction accuracy is being optimised and it is not well suited to interpretability. Why then constrain the variable list so heavily before applying RF?

The large number of variables included could imply a priori that some of them share information. In anticipation of this and to avoid confusion between variables finally included in the machine learning models, logistic regression was performed as the first step of the multivariate analysis, as described in Methods, in the Multivariate analysis section. We have slightly modified its wording to reinforce the idea.

On the interpretation of results, my takeaway is that three of the four final predictors of moderate-severe disease at follow-up are markers of more severe disease at baseline (PsAID, tender joint count, and physical activity which could be a consequence of disease level). I’m not sure how beneficial this is? Have the authors considered change in disease activity as an outcome?

According to the work previously mentioned (reference 19 in our article), DAPSA percentage change is used to define treatment response. Though it can be an interesting endpoint, it was not the aim of our work to analyse predictors of treatment response, but predictors of higher disease activity.

Part of the interest of our work would be that there is no confounding effect affecting the associations found between the explanatory variables and moderate-high disease activity. According to this, the four final predictors have an association with disease activity once the effect of the rest of variables initially selected in the bivariate analysis has been taken into account. Moreover the association of each of the four predictors with disease activity in the following annual visit would be independent between them. This would reflect an association that would go, at least in part, beyond disease activity at the present moment. We have added this in the Discussion, before the Conclusions.

It is unclear whether treatments are included as potential predictors with variable categories for individual drugs or just for class of drug. I think it’s important to give some more detail about treatment in particular so the reader can understand whether the study is predicting prognosis or treatment response. The end of the discussion states “The main strength of this study is its ability to record the course of PsA from an early phase before the natural disease evolution is modified by treatment prescribed by the rheumatologist” but presumably the effect of treatment is present in the outcome.

We considered treatment of PsA with DMARDs, differentiating synthetic and biologic DMARDs, but not individual drugs; we have specified it in the Methods section, when describing variables and measures. What the end of the discussion means is that the baseline visit reflected the situation of the patients before the evolution was modified by treatment prescribed by the rheumatologist. We have slightly modified its wording to reinforce the idea. 

The authors have apparently previously published a severity prediction study in the same dataset (reference 6). This should be described more clearly in the introduction and it should be made clear what is the novel additional contribution of the current manuscript. Also worth explaining DAPSA in the introduction.

Thank you very much for this suggestion. Indeed, in a previous study we analyzed which patient and disease characteristics could be useful to predict a more severe disease among recent-onset psoriatic arthritis patients. However, the concept of severity should not be made homonymous with that of inflammatory load, which is essentially what the DAPSA captures. The severity of a disease brings together not only the inflammatory load of the disease, but also physical disability, structural damage, and the impact on quality of life, among others. The factor that perhaps weighs the most in this equation, especially at the beginning of the disease, is inflammatory activity, but with the passage of time this loses its predictive capacity, and there are other aspects (structural damage, comorbidities, deterioration in physical function, etc) more clearly associated with greater severity. In fact, our two studies (the current one and the one referring to severity) point to different factors in the prediction of both outcomes, which gives greater strength to our thesis that both aspects (inflammatory load and severity) are different constructs of the disease. In addition, more than one study has shown how the outcome measures on disease activity and those on the negative impact on QoL are clearly discordant, which suggests that both aspects measure different things within the overall health of patients with PsA.

As for the DAPSA, a brief description of it is made in the introduction. 

Minor comments:

** Introduction **

Line 47 – it would be beneficial to expand on what is meant by multidimensional instruments

Thanks again. A brief description is made about it

** Methods **

More specific detail should be given. For example:

-       A supplementary table listing treatment numbers.

            We have added the numbers for prescription of synthetic and biologic DMARDs in the baseline visit, at the beginning of the Results.

-       Summary of the times between visits (i.e. prediction interval)

This work is part of the REAPSER study, a multicenter observational prospective study with 2-year follow-up period and regular annual visits. The analysis aimed to estimate predictive ability, seeking associations between the outcome measure (moderate-high activity of PsA) and values at the previous visit for the remaining variables. This is written when explaining the generation of the dataset (point 2.3.2. in Materials and Methods).

It is not clear how the initiation and finalisation dates of treatment (variable m) are handled if treatment is not stopped. Also does this include historical treatments?

Initiation and finalization dates were only considered to define if the patient was in treatment with synthetic or biologic DMARDs in the different visits. Considering that this could lead to missunderstanding, we have deleted it in the description of variable m.

Regarding historical treatments, baseline visit evaluated the patient’s situation before disease progress was modified by the treatments prescribed by the rheumatologist (second paragraph of Materials and Methods). Therefore, there were no historical treatments for PsA.

It would be useful to give the total number of predictive features considered as the list is very long and it is unclear.

Following reviewer´s advice, we have added the total number at the beginning of 2.1.Variables and measurement.

Also, line 131 – presumably variable (n) is the outcome variable. It should be clearer that this is the outcome and treated as a binary outcome. Or is it also included as a predictor at baseline?

It is the outcome and treated as a binary outcome. Following reviewer´s advice, we have added this information in this item (n) to make it clearer.

Line 165 – should be XGBoost algorithm “with” the SHAP technique? (rather than “and”), i.e. one approach is being described not two? You should also be specific about the inclusion criteria for informative variables that is being applied with this approach.

We appreciate reviewer´s comment. We have written “with” instead of “and”. We have also specified in 2.3.3. Bivariate analysis and in Appendix A the inclusion criteria applied with this approach.

What does SHAP stand for?

It stands for SHapley Additive exPlanations. We have added it in Appendix A

Line 173 – 75% training dataset – worth describing in section 2.3.2 what this is and how it is generated e.g. is it stratified/balanced in any way? Line 177 – “sample is split” – is this the same 75/25% split? It needs to be clearer.  

As currently presented the 25% test dataset does not appear to be used anywhere. Please clarify

Following reviewer´s advice, we have added this description in the second and third paragraph of 2.3.4. Multivariate analysis. 

** Results **

Line 200 - 59.6% of pts have moderate-high disease activity – this is insufficiently clear in the context of multiple outcome measures per patient.

This refers to the baseline visit. On the other hand, 20.8% and 21.2% of the patients who attended the clinic had moderate-high disease activity at the first and second follow-up visit, respectively. This is specified at the beginning of the Results. 

Table 1 – what do table values represent? Count (%) and mean (range) or median (IQR) etc? It would benefit to add the total n for the two columns. Also perhaps indicating via footnotes which tests were performed to generate the p-values

Categorical variables are expressed as n (%) and numeric variables as median [interquartilic range]. Tests performed to generate the p-values were the Mann-Whitney test for numeric variables and the Chi-square test for categorical variables. We have added this information and the total n for the two columns in Table 1.

Line 211 – 319 observations in total. Please clarify how this can be more than 158 patients x 2 visits = 316

It is due to the oversampling technique: When the subsamples are imbalanced, those data whose value for the outcome variable is a minority value are duplicated or triplicated to train the models. This is written in the third paragraph of 2.3.4. Multivariate analysis. 

Table 3 – I feel this can be summarised more clearly as a figure or by giving total/average importance measures

We appreciate the reviewer´s comment. We would like to keep the table as we consider it is more informative of the analysis.

** Discussion **

Lines 259-263 – I’m not clear what point is being made. Perhaps worth reminding the reader of the differences between PsAID and DAPSA?

PsAID is an instrument to capture the impact of the disease on the quality of life of PsA patients, while DAPSA is a tool to assess disease activity in clinical routine. We have added an explanatory paragraph.

Line 267 – “many patients with PsA and no obvious arthritis in the physical examination” – I’m not a clinician so I’m not clear what this means – psoriatic arthritis without arthritis?

This alludes to the fact that in clinical practice it is not uncommon to find patients with a correct diagnosis of the disease, in whom there are apparently no signs of inflammation on physical examination, but nevertheless these signs may be present on imaging techniques such as ultrasound or MRI (this is now called subclinical disease).

Line 275 – “have some subclinical inflammatory activity that can only be detected using sensitive imaging techniques.” – how do you infer this when your final model is based on tender joint count and not imaging data?

Thank you for this comment. What we are referring to in the discussion when addressing this point, and consistent with the previous point, is that patients with joint pain without apparent signs of inflammation on physical examination, may in fact have such inflammatory activity if we subjected them to more sensitive imaging studies. However, it remains speculative reasoning.

Physical activity levels are discussed as a causal factor of activity levels and not considered a potential consequence.

Taking into account that the association between physical activity levels and disease activity on the following year´s visit is observed after adjusting by the tender joint count at the present moment in the logistic regression analysis, we consider that this association would have a causal component.

Sex as a predictor is interesting. However, it’s not clear whether this just represents “high activity predicts high activity” again, depending on whether females have higher activity at baseline. Could this test be done at baseline? I also feel that an opportunity is missed to explore further the impact of sex in relation to the other predictors, e.g. looking at potential interactions between sex and the severity-related measures in prediction of future activity.

We appreciate the reviewer´s comment. Similarly to the previous answer, the association between sex and disease activity on the following year´s visit is observed after adjusting by the tender joint count and PsAID in the logistic regression analysis, pointing out that this association would go beyond different disease activity or impact in the previous visit. We have added this in the Discussion

Limitations don’t mention the lack of independent replication which I think is also key.

Following reviewer´s advice, we have added this limitation in the Discussion.

** Appendix A **

Line 395 – “the null hypothesis of this test is that there would be no statistically significant differences in the functioning of the model if the coefficient of the variable was zero” – a null hypothesis can’t refer to statistically significant differences. The null hypothesis is just that the effect of a variable is zero.

We have corrected it as indicated by the reviewer.

Is the logistic regression approach referred to simply stepwise backward regression? Why not state this?

It is stepwise backward regression. We have added it in the Appendix.

Reviewer 2 Report

I read with interest the manuscript by Queiro et al. The manuscript is well-written and the English is fine. 

I suggest the author to better address what is in branches in the tables (%, ranges, standard deviation) and to state a comment on the utility of these technology in clinical practice. 

Author Response

I suggest the author to better address what is in branches in the tables (%, ranges, standard deviation) and to state a comment on the utility of these technology in clinical practice. 

We appreciate the reviewer´s comment. We have added the information in Table 1. We have also completed the Discussion about the utility.